# Hyperbaric Oxygen Therapy Reduces Oxidative Stress and Inflammation, and Increases Growth Factors Favouring the Healing Process of Diabetic Wounds

**DOI:** 10.3390/ijms24087040

**Published:** 2023-04-11

**Authors:** Xavier Capó, Margalida Monserrat-Mesquida, Magdalena Quetglas-Llabrés, Juan M. Batle, Josep A. Tur, Antoni Pons, Antoni Sureda, Silvia Tejada

**Affiliations:** 1Research Group in Community Nutrition and Oxidative Stress, University of the Balearic Islands—IUNICS, 07122 Palma, Spain; xavier.capo@ibsalut.es (X.C.); margalida.monserrat@uib.es (M.M.-M.); m.quetglas@uib.es (M.Q.-L.); info@medisub.org (J.M.B.); pep.tur@uib.es (J.A.T.); antonipons@uib.es (A.P.); 2Translational Research in Aging and Longevity (TRIAL) Group, Health Research Institute of the Balearic Islands (IdISBa), 07120 Palma, Spain; 3Health Research Institute of Balearic Islands (IdISBa), 07120 Palma, Spain; silvia.tejada@uib.es; 4CIBER Fisiopatología de la Obesidad y Nutrición (CIBEROBN), Instituto de Salud Carlos III (ISCIII), 28029 Madrid, Spain; 5MEDISUB Recerca, 07400 Alcúdia, Spain; 6Laboratory of Neurophysiology, Department of Biology, University of the Balearic Islands, 07122 Palma, Spain

**Keywords:** hyperbaric oxygen therapy, inflammation, wound healing, oxidative stress

## Abstract

Hyperbaric oxygen therapy (HBOT) is the clinical application of oxygen at pressures higher than atmospheric pressure. HBOT has been effectively used to manage diverse clinical pathologies, such as non-healing diabetic ulcers. The aim of the present study was to analyse the effects of HBOT on the plasma oxidative and inflammation biomarkers and growth factors in patients with chronic diabetic wounds. The participants received 20 HBOT sessions (five sessions/week), and blood samples were obtained at sessions 1, 5 and 20, before and 2 h after the HBOT. An additional (control) blood sample was collected 28 days after wound recovery. No significant differences were evident in haematological parameters, whereas the biochemical parameters progressively decreased, which was significant for creatine phosphokinase (CPK) and aspartate aminotransferase (AST). The pro-inflammatory mediators, tumour necrosis factor alpha (TNF-α) and interleukin 1β (IL-1β), progressively decreased throughout the treatments. Biomarkers of oxidative stress––plasma protein levels of catalase, extracellular superoxide dismutase, myeloperoxidase, xanthine oxidase, malondialdehyde (MDA) levels and protein carbonyls––were reduced in accordance with wound healing. Plasma levels of growth factors––platelet-derived growth factor (PDFG), transforming growth factor β (TGF-β) and hypoxia-inducible factor 1-alpha (HIF-1α)–– were increased as a consequence of HBOT and reduced 28 days after complete wound healing, whereas matrix metallopeptidase 9 (MMP9) progressively decreased with the HBOT. In conclusion, HBOT reduced oxidative and pro-inflammatory mediators, and may participate in activating healing, angiogenesis and vascular tone regulation by increasing the release of growth factors.

## 1. Introduction

Hyperbaric oxygen therapy (HBOT) is based on the utilisation of oxygen at pressures higher than atmospheric, commonly at 2–3 atmospheres with 100% oxygen exposure [1,2]. HBOT increases oxygen availability for body tissues, including plasma, and increases the capacity of blood to transport oxygen with respect to the concentration of normobaric conditions, which is often associated with pharmacological effects [3]. Under physiological conditions, haemoglobin is 97% saturated; thus, increasing haemoglobin saturation does not lead to a substantial improvement in tissue oxygen supply [4]. However, the concentration of dissolved oxygen in the plasma can be significantly increased by hyperbaric therapy. Therefore, according to Henry’s Law, an increase in pressure will translate into a greater oxygen solution that can be transported to tissues.

Chronic wounds are defined as those that have failed to proceed through an orderly and timely series of events to produce a durable structural and functional closure [5]. The main characteristic of chronic wounds is the low partial pressure of oxygen at the centre of the wound, which hinders healing [6]. For this reason, HBOT has been used as treatment against several pathologies characterised by tissue hypoxia, such as diabetic wounds, carbon monoxide poisoning, gas, gangrene, necrotising fasciitis, compartment syndrome, intracranial abscesses, burns and a sequelae of radiation treatments or osteomyelitis [7,8]. In this sense, it has been demonstrated that diabetic foot wounds progress favourably after treatment with HBOT [1,6]. Moreover, HBOT preconditioning for surgery has also been reported to reduce complications and hospital stays [9]. In addition, the fact that HBOT increases dissolved oxygen may favour its use in photodynamic therapy to reduce cancer progression [10].

Diabetic foot ulcers or wounds, are one of the most frequent and serious complications associated with diabetes, affecting about 15% of all patients and accounting for about 50% of all lower limb amputations [11,12,13]. A diabetic foot ulcer results from a combination of causes, the main underlying ones being peripheral neuropathy and ischemia from peripheral vascular disease [14,15]. Diabetic foot wounds are characterised by decreased angiogenesis, ischemia, persistent inflammation and reduced antioxidant defences that interfere with endogenous healing mechanisms, causing the wound to be in a chronic inflammatory state without progressing to the resolution phase [6,16,17]. Wound healing is a multifactorial process outlined by three phases (inflammation, proliferation and remodelling) and involves the participation of several cell types, such as fibroblasts, immune cells, keratinocytes and endothelial cells [16,18]. As consequence of the skin damage, several compounds with chemoattractant properties are secreted to the circulation, leading to the migration and recruitment of neutrophils (at the beginning) and macrophages (as late response). Recruited leukocytes phagocytose necrotic tissues and release cytokines, such as tumour necrosis factor-α (TNF-α), interleukin (IL)-1β and IL-6, and growth factors, such as vascular endothelial growth factor (VEGF) and insulin-like growth factor-1 (IGF-1), that mediate and facilitate healing [18,19]. Due to the complexity of the wound healing process, their treatment should be approached in different ways, including reducing inflammation, extensive local wound care, revascularization of ischemic extremities, reduction of infections and the improvement of peripheral circulation [20,21].

Oxygen plays a central role in wound healing. It is not only necessary for cell respiration, but also as a source of reactive oxygen species (ROSs), which are necessary in several physiological processes, such as cell communication, bactericidal activity or angiogenesis promotion [18,22]. In addition, low subcutaneous oxygen tension is related to a higher infection risk [23]. In this sense, HBOT induces the production of ROSs, which are believed to destroy invading bacteria and protect against infection [24]. Other authors have reported that HBOT can reduce tissue oedema, inflammation and oxidative stress [9,25]. In addition, HBOT can induce an increase in the expression of several growth factors, such as hypoxia-inducible factor 1 (HIF-1), which could induce angiogenesis and cell proliferation [9,26,27]. A previous study revealed that HBOT induced angiogenesis, regulated vascular tone, improved antioxidant plasma status and promoted the resolution of inflammation [1]. Other studies showed that HBOT induced fibroblast replication, osteoclast activation, and the upregulation of VEGF and platelet-derived growth factor (PDGF), which together contribute to wound healing [28]. However, several authors have not found any positive effects of HBOT in wound healing [29,30]. Moreover, others even reported adverse effects, such as barotraumas, generalized seizures, transient myopia and pulmonary toxicity, although at very low frequencies [31,32,33]. Taking this into account, the objective of this work was to evaluate the mechanisms involved in the improvement of the diabetic foot in patients undergoing HBOT, using biomarkers of oxidative stress, inflammation and growth factors.

## 2. Results

All included patients used conventional treatments to heal chronic wounds that failed to respond. At the beginning of the study, 21 patients were included, but two of them voluntarily left the study before its completion and one did not respond to therapy. Total wound healing was not achieved after treatment with HBOT; therefore, it was not included in the final data analysis. Changes in haematological and biochemical parameters before sessions 1, 5, 20 and 28 (days) after the HBOT are presented in Table 1.

No HBOT effects were evident in blood cells, haemoglobin or the haematocrit. A progressive decrease in biochemical parameters was observed throughout the treatment, although the differences were only significant for creatine phosphokinase (CPK) and aspartate aminotransferase (AST) (*p* < 0.05).

Table 2 shows catalase (CAT), extracellular superoxide dismutase (EcSOD), myeloperoxidase (MPO) and xanthine oxidase (XOX) plasma levels (%) after sessions 1, 5, 20 and 28 (days) after the HBOT. A progressive, but insignificant, decrease in CAT protein plasma levels was evident throughout the HBOT. However, lower CAT protein levels were found 28 days after wound recovery. EcSOD plasma levels decreased during the treatment and continued decreasing 28 days after wound recovery, although this decrease was not significant. MPO plasma levels decreased progressively and significantly after each HBOT, with the lowest levels 28 days after wound recovery. XOX plasma levels followed a pattern of response very similar to that of MPO. An important decrease in XOX plasma levels was observed after five HBOT sessions, but no differences between XOX plasma levels after five and 20 HBOT sessions were evident. In contrast, a significant and lower XOX plasma level 28 days after wound recovery was observed.

MDA and carbonyl derivate levels, as markers of oxidative damage, are also presented in Table 2. Both parameters significantly reduced their levels during the HBOT, with lower levels 28 days after wound recovery. In relation to MDA, the levels significantly decreased after days 5 and 20 of HBOT, with no differences between them. The lowest MDA plasma levels were evident 28 days after wound recovery. Carbonyl derivate levels significantly decreased after 20 days of treatment, and these levels were maintained 28 days after wound recovery.

TNF-α and IL-1β, as markers of the inflammatory status, are presented in Figure 1. Both cytokines showed a similar response pattern. 

TNF-α plasma levels were reduced by almost half after 5 days of treatment, and this reduction continued progressively throughout HBO treatment, with the significantly lowest level 28 days after wound recovery. In a similar manner, IL-1β plasma levels also reduced progressively throughout the HBOT, with no differences between the first and fifth HBOT. IL-1β lowest plasma levels were evident after 20 days of HBOT and 28 days after wound recovery, without significant differences between them.

TGFβ, PDGF and HIF-1α plasma levels are presented in Figure 2. PDGF and HIF-1α showed a similar response, with a progressive increase in plasma levels. In both cases the levels increased after 5 days of HBOT, but only in the case of HIF-1α was the increase statistically significant. After 20 days of HBOT, both PDGF and HIF-1α presented higher plasma levels. Both PDGF and HIF-1α showed a slight, but not significant, decrease in their levels 28 days after wound recovery. The TGFβ levels significantly increased after 5 days of HBOT and decreased progressively until the end of the treatment, when the lowest plasma levels were observed.

Figure 3 represents MMP9 plasma level variations following sessions 1, 5, 20 and 28 (days) after wound recovery. These levels progressively decreased during HBOT, but this decrease was only significant 28 days after wound recovery. The plasma levels of MMP9 28 days after wound recovery were significantly lower when compared with the beginning of the study and after 5 days of HBOT. 

## 3. Discussion

The skin layer can suffer alterations that lead to wound healing, a process that implies inflammation, angiogenesis and extracellular remodelling to repair the damage [34,35], such that evaluation of different markers related to this phase can give information related to cell recovery. In the current work, a considerable reduction in CPK and AST activity throughout the HBOT was observed, which was parallel to improvements in wound healing. Similarly, it had been previously shown that skin damage induced an increase of up to three times in CK levels [36]. Patients with a severe skin injury, such as toxic epidermal necrolysis or second- or third-degree burns, also showed increased CK plasma activity [37,38]. Additionally, it has been shown that elevated levels of liver enzymes, such as ALT and AST, are associated with slower wound healing, probably as a result of the relationship between diabetes and a fatty liver, which can worsen the metabolic complications of diabetes [37].

It is generally accepted that chronic wounds are directly related to the oxidative stress status [39,40]. In fact, evidence of high ROS levels in chronic and non-healing wounds has previously been described [41]. ROSs play a dual role in wound healing: on one hand, a high ROS concentration is necessary to reduce the risk of infection in the wound area, but several other studies have reported that low ROS amounts are also necessary for wound healing by acting as cellular signallers [42,43]. However, a highly unregulated ROS production can damage biomolecules, such as lipids, DNA and proteins, thereby causing oxidative stress [44]. In this sense, high plasma levels of oxidative damage markers (MDA and carbonyl protein derivates) were observed at the beginning of the treatment in the current work, which was followed by a progressive reduction throughout the sessions. These initial amounts of oxidative stress markers were probably related to a high ROS production as consequence of the chronic wound. In addition, previous studies reported that HBOT reduced ROS levels, conferring an increase in antioxidant plasma capabilities throughout [1,45]. The later reduction in oxidative damage markers coincided with an acute decrease in XOX plasma levels. XOX, mainly superoxide anion as consequence of xanthine or hypoxanthine oxidation, is an important ROS source [46]. In addition, high XOX is often associated with several pathologies, such as diabetes, atherosclerosis, hypertension, endothelial dysfunction and inflammation [47,48]. Concretely, high amounts of ROSs in chronic wounds induce a proinflammatory cytokine secretion and MMP activation [40]. In the present study, the reduction in the levels of XOX and MPO (pro-oxidant and pro-inflammatory enzymes) coincided with a reduction in TNF-α, IL-1β and MMP9 levels in plasma. At the same time as oxidative damage (MDA and carbonyl protein derivates) being observed, a progressive decrease in CAT and EcSOD plasma levels was detected. These two facts reinforced the link between oxidative stress, inflammation and wound healing. Accordingly, a transient increase in nuclear factor erythroid 2-related factor 2 (Nrf2) expression was observed after an HBOT session, recovering basal levels after 24 h [12]. However, this increase was attenuated when HBOT was continued in a clinical procedure, This could have been related to a lower degree of oxidative stress and a lower expression of antioxidant enzymes as the wound improved [49].

Inflammation is the first step in wound healing [50], during which leukocytes release pro-inflammatory cytokines, such as TNF-α, IL-1β and IL6 [51]. It is generally known that TNF-α and IL-1β levels are high in chronic wounds [51]. This was in accordance with the highest TNF-α and IL-1β plasma levels found at the beginning of the current study, but these levels decreased progressively with HBOT sessions, indicating an improvement in inflammation throughout the wound healing process. In addition, previous studies have indicated that pro-inflammatory cytokines, mainly TNF-α, IL6, IL8 and IL-1β, remained high in the non-healing phase, while their levels were reduced when the healing process occurred [52]. The anti-inflammatory activity associated with HBOT seemed to be mediated by the inhibition of pro-inflammatory nuclear factor κB (NF-κB) [9]. Under hypoxic conditions, the protein IκBα that binds to NF-κB and maintains it in an inactive state in the cytoplasm was degraded [53]. Thus, the hyperoxia produced during HBOT could promote the preservation of IκBα, inhibiting the activation of the NF-κB signalling pathway and, consequently, the transcription of pro-inflammatory cytokines [54].

Growth factors are bioactive peptides that act in coordination with cytokines, MMPs, inflammatory cells or other cell types, such as keratinocytes or fibroblasts, to affect wound healing [52]. However, in chronic diseases, alterations to some growth factors are evident, especially a reduced production, which delays wound healing [53]. PDGF plays a key role in the healing process by inducing the proliferation of fibroblasts and the production of the extracellular matrix, which favours the recovery of connective tissue [54]. Similarly, TGFβ promotes the stimulation of inflammatory cells, keratinocytes and fibroblasts, favouring vascularization, angiogenesis and the formation of the extracellular matrix [55]. The expression of both TGFβ, PDGF and their receptors has been shown to be decreased in diabetic wounds [56,57]. In this sense, it is also well established that growth factors, including PDGF, TGFβ and VEGF, increase in plasma levels during healing [55]. HBOT induced an increase in PDGF and TGFβ after 20 sessions, which indicates wound healing. Moreover, a progressive increase in HIF-1α in the plasma of patients was observed until day 20 of treatment. In chronic wounds, as a consequence of low partial pressure of oxygen at the centre of the wound [6], there was an activation of HIF-1α. This factor regulates angiogenesis during wound healing [56], since it induces the expression of VEGF [57], which, in turn, stimulates the formation of blood vessels. In addition, HIF-1α can also modulate PDGF levels [58], contributing to skin repair. In this sense, it has been shown that the expression and release of growth factors and angiogenesis mediators is regulated by NF-κB and HIFα, so their response to HBOT can lead to tissue repair, inflammation reduction and the release of growth factors [59]. Specifically, HBOT has been reported to increase HIF-1α stability against degradation, followed by an increase in its transactivation [27]. 

MMP-9 is an extracellular matrix remodelling protein involved in the wound healing that is transiently expressed during normal wound healing, although it has been found at high levels in chronic wounds [60,61]. Usually, high MMP9 levels are associated with a pro-inflammatory status [60]. In this sense, some studies proposed that high amounts of MMP9 are harmful for wound healing. In fact, there is evidence that inhibiting MMP9 accelerated wound healing [62]. The current results showed a progressive reduction in MMP9 levels as a consequence of HBOT 28 days after wound recovery. For instance, HBOT was able to inhibit the activity and expression of MMP9 in the heart of diabetic rats [63]. In this sense, the increase in ROS induced by HBOT downregulated the mitogen-activated protein kinase (MAPK) pathway, thus decreasing the expression of various MMPs, such as MMP9, which favours the deposition of the extracellular matrix [64]. Moreover, TNFα was also shown to regulate MMP9 expression [61]. 

In conclusion, HBOT could be an important tool for treating chronic wounds that are not healed by conventional treatments. Because it induces improved healing, the treatment could lead to a higher oxygen disposal during the sessions, which would allow for a quicker resolution of wound inflammation and the onset of angiogenesis through the activation of several growth factors. As treatment progresses, there is an increase in antioxidant enzymes (CAT and EcSOD) and a decrease in pro-oxidant enzymes (MPO and XOX), inflammation markers (TNF-α and IL-1β) and the MMP9. In the case of growth factors (TGFβ, PDGF and HIF-1α), there is an increase with HBOT, tending to return to initial values after complete wound healing. Although the mechanisms involved in HBOT-induced wound healing are dynamic and complex, some signalling pathways, such as NF-κB, HIF-1 α and MAPKs, seem to play an important role. Future studies to increase understanding of these pathways may provide information to improve diabetic wound management with HBOT.

## 4. Materials and Methods

### 4.1. Patient Characteristics

A total of 18 patients (65.7 ± 5.5 years) who had chronic non-healing diabetic wounds voluntarily participated in the study. The main inclusion criteria included no response to previous conventional treatments: topical antibiotics, topical dressings and debridement of tissue. In addition, participants had to be non-smokers and to not have taken any antioxidant dietary supplement at least for one month before the study. During the HBOT, wounds were cleansed with saline, treated with antibiotics and periodically debrided of necrotic tissue to obtain a well-bleeding granulation base. The experimental procedure was designed in accordance with the recommendations for clinical research by the Declaration of Helsinki, and was approved by the Ethical Committee of Clinical Investigation of the Government of Balearic Islands (Palma de Mallorca, Balearic Islands, Spain), with approval number IB1295/09PI. All participants were informed of the purpose and the implications of the study, and all provided written consent to participate.

### 4.2. Experimental Procedure

The patients were exposed to 20 HBOT sessions (Monday to Friday, for four weeks) in a hyperbaric chamber and breathing 100% oxygen, at a pressure of 2.2 ATA for 1 h. Blood samples were obtained from the antecubital vein before the first, fifth and twentieth sessions. In addition, a supplementary sample was collected 28 days after wound recovery, to be used as a control sample (Figure 4). Haematological parameters were determined using an automatic flow cytometer analyser Technicon H2 (Bayer, Frankfurt, Germany), and clinical biochemical parameters were measured in the serum by standard procedures using the auto-analyser Technicon DAX System (Bayer, Frankfurt, Germany). Plasma was obtained after centrifugation of the blood samples collected with EDTA, at 900× *g* and 4 °C for 30 min. Plasma samples were immediately stored at −80 °C and all biochemical procedures were carried out in duplicate.

### 4.3. Western Blot Analysis

CAT, EcSOD, MPO and XOX protein levels in plasma were determined by Western blot (Appendix A). Plasma samples were analysed by SDS-PAGE (Bio-Rad Laboratories, Alcobendas, Madrid, Spain). Thirty micrograms of the total protein were loaded on a 12% PAGE. Following electrophoresis, samples were transferred onto a nitrocellulose membrane and incubated with a primary monoclonal anti-CAT antibody (Ref: 219010, Calbiochem; Bionova, Madrid, Spain), anti-MPO antibody (Ref: sc-52707 Santa Cruz Biotechnology, Heidelberg, Germany), anti-XOX antibody (Ref: 3604-3-250, Mabtech AB, Nacka Strand, Sweden) or anti-EcSOD antibody (Ref: SOD-106, Assay Designs Inc., Ann Arbor, MI, USA), and the corresponding secondary anti-rabbit IgG peroxidase-conjugated antibody. Protein bands were visualised using an enhanced chemiluminescence kit, Immun-StarR^©^ Western CR^©^ Kit (Bio-Rad Laboratories, Alcobendas, Madrid, Spain). The chemiluminescence signal was captured with the Chemidoc XRS densitometer (Bio-Rad Laboratories, Alcobendas, Madrid, Spain), and analysed with the Quantity One Software (Bio-Rad Laboratories, Hercules, CA, USA). Tubulin (Ref: sc-23948 Santa Cruz Biotechnology, Heidelberg, Germany) was used as a housekeeper protein to normalise the Western blot bands for protein loading.

### 4.4. MDA Levels

As a marker of lipid peroxidation, MDA concentration (μM) was analysed in the plasma and in PBMCs of all participants by a specific colorimetric assay kit (Merck Life Science S.L.U., Madrid, Spain), following the manufacturer’s instructions [65]. Absorbance was measured at 586 nm in a microplate reader (Bio-Tek^®^ PowerWaveXS, Agilent Technologies, Madrid, Spain).

### 4.5. Protein Carbonyl Derivates

Plasma protein carbonyl derivates (10 mg of protein) were determined by an immunological method using the OxiSelect™ Protein Carbonyl Immunoblot Kit (Cell Biolabs, San Jose, CA, USA), following the manufacturer’s instructions. Total protein concentrations were measured by the Bradford method. Samples were transferred to a nitrocellulose membrane by the dot-blot method. Image analysis was performed using the Quantity One-1D analysis software (version 4.6.5, Bio-Rad Laboratories, Hercules, CA, USA).

### 4.6. ELISA Assays

TGFβ, PDGF, HIF-1α, TNF-α, IL-1β and MMP9 plasma levels were measured using commercial ELISA assays kits.

TGFβ, PDGF and HIF-1α were purchased from Elabscience Biotechnology Inc^®^ (Houston, TX, USA). Intra-assay and inter-assay reproducibility for TGFβ was lower by 5 and 5.5%, respectively. In the case of PDFG, it was lower by 6 and 5.5%, respectively. HIF-1α intra-assay and inter-assay reproducibility was lower by 4.5 and 5%, respectively. 

TNF-α and IL-1β ELISA kits were purchased from DIACLONE^®^ (Besançon, France). Intra-assay and inter-assay reproducibility for TNF-α was 3.2 and 10.9%, respectively. For IL-1β, it was 4.5 and 8.7%, respectively.

The MMP9 ELISA kit was purchased from Cloud-Clone Corp^®^ (CCC, Houston, TX, USA). Intra-assay and inter-assay reproducibility was lower by 10 and 12%, respectively.

### 4.7. Statistical Analysis

The SPSS statistical software package, version 25.0 (SPSS Inc., Chicago, IL, USA) was used for the statistical analysis. The normal distribution of the data was assessed by the means of the Kolmogorov–Smirnov test. The statistical differences between the obtained data were evaluated by a one-way analysis of variance (ANOVA). When significant differences were reported, a Bonferroni post hoc test was used to establish the differences between the different data. Data are presented as the mean ± SEM, and *p* < 0.05 was considered to be statistically significant. 

## Figures and Tables

**Figure 1 ijms-24-07040-f001:**
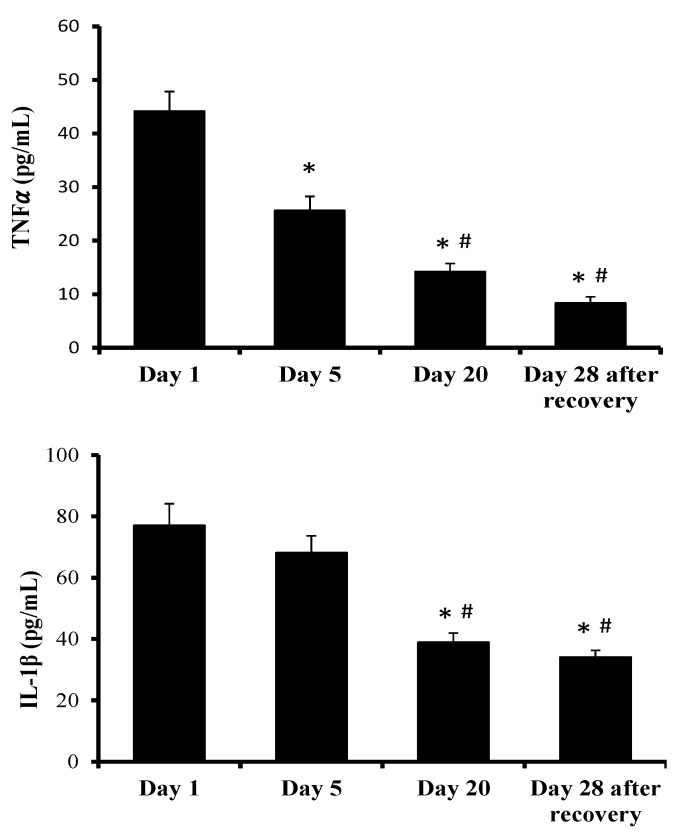
TNF-α and IL-1β plasma levels following HBOT sessions 1, 5, 20 and 28 (days) after wound recovery. The effects of the HBOT sessions were evaluated by a one-way ANOVA. *p* < 0.05; * indicates significant differences with respect to session 1; ^#^ indicates significant differences with respect to session 5. Abbreviations: TNF-α, tumour necrosis factor-alpha; IL-1β, interleukin-1 beta.

**Figure 2 ijms-24-07040-f002:**
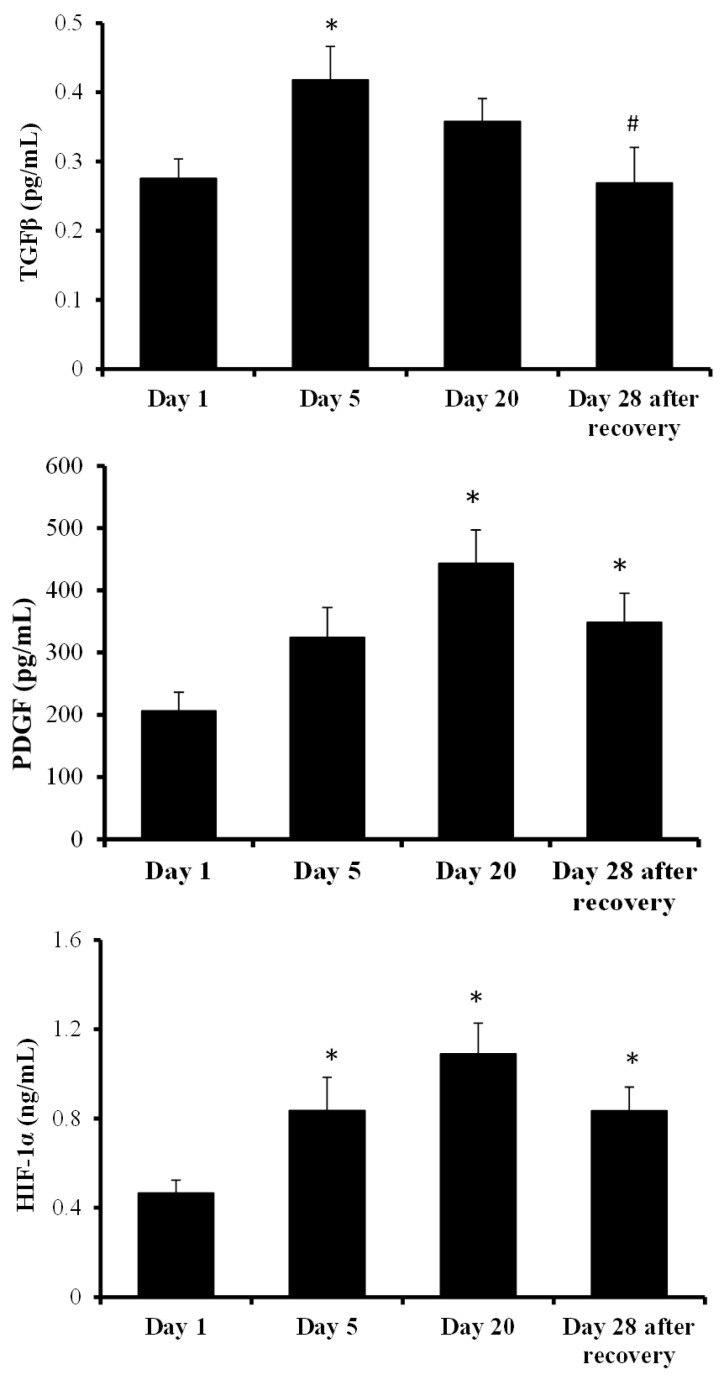
TGFβ, PDGF and HIF-1α plasma levels following HBOT sessions 1, 5, 20 and 28 (days) after wound recovery. The effects of the HBOT sessions were evaluated by a one-way ANOVA. *p* < 0.05; * indicates significant differences with respect to session 1; ^#^ indicates significant differences with respect to session 5. Abbreviations: TGFβ, transforming growth factor beta; PDGF, platelet-derived growth factor; HIF-1α, hypoxia-inducible factor 1.

**Figure 3 ijms-24-07040-f003:**
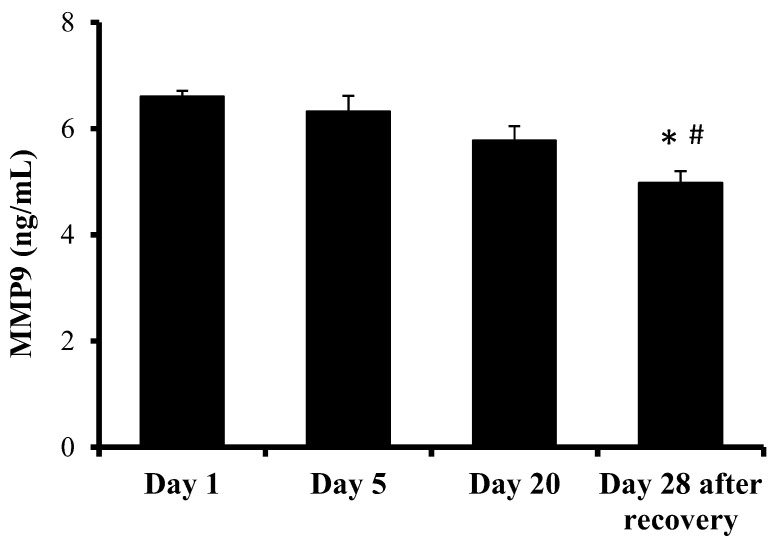
MMP9 plasma levels following HBOT sessions 1, 5, 20 and 28 (days) after wound recovery. The effects of the HBOT sessions were evaluated by a one-way ANOVA. *p* < 0.05; * indicates significant differences with respect to session 1; ^#^ indicates significant differences with respect to session 5. Abbreviation: MMP9, matrix metallopeptidase 9.

**Figure 4 ijms-24-07040-f004:**
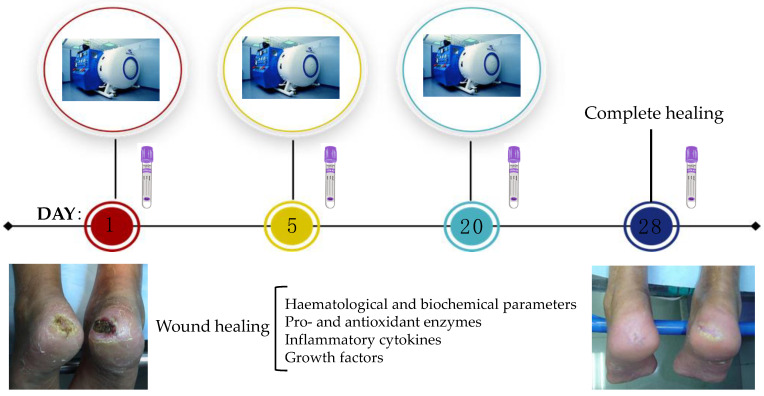
Scheme of the experimental procedure and the main parameters determined during the wound healing process.

**Table 1 ijms-24-07040-t001:** Haematological and serum biochemical parameters before sessions of the HBOT, 1, 5, 20 and 28 (days) after wound recovery.

	Day 1	Day 5	Day 20	Day 28
Erythrocytes (10^6^ cells/µL)	4.16 ± 0.08	4.12 ± 0.07	4.13 ± 0.08	4.21 ± 0.10
Haematocrit (%)	38.8 ± 0.6	38.1 ± 0.6	38.4 ± 0.5	39.1 ± 0.7
Haemoglobin (g/L)	12.8 ± 0.9	12.6 ± 0.9	12.7 ± 1.0	12.8 ± 0.8
Leukocytes (10^3^ cells/µL)	5.97 ± 0.18	6.18 ± 0.26	6.24 ± 0.23	5.89 ± 0.28
CPK (U/L)	256 ± 26	178 ± 17 *	124 ± 10 *^,#^	122 ± 13 *^,#^
LDH (U/L)	370 ± 15	352 ± 17	340 ± 19	335 ± 19
AST (U/L)	28.9 ± 2.2	26.7 ± 2.5	22.7 ± 1.1*	22.1 ± 1.3 *
ALT (U/L)	29.6 ± 4.5	28.9 ± 4.6	23.0 ± 2.5	22.9 ± 2.8
GGT (U/L)	57.1 ± 5.0	53.2 ± 5.7	46.6 ± 4.9	40.7 ± 7.4
Creatinine (mg/dL)	1.07 ± 0.4	1.04 ± 0.04	1.00 ± 0.04	0.98 ± 0.03

The effects of HBOT sessions were evaluated by a one-way ANOVA. *p* < 0.05; * indicates significant differences with respect to session 1; ^#^ indicates significant differences with respect to session 5. Abbreviations: CPK, creatine phosphokinase; LDH, lactate dehydrogenase; AST, aspartate aminotransferase; ALT, alanine aminotransferase; GGT, gamma-glutamyltransferase.

**Table 2 ijms-24-07040-t002:** CAT, EcSOD, MPO and XOX plasma protein levels (%) following sessions 1, 5, 20 and 28 (days) after wound recovery.

	Day 1	Day 5	Day 20	Day 28
CAT (%)	1.80 ± 0.19	1.70 ± 0.21	1.66 ± 0.10	1.00 ± 0.09 *^,#,&^
EcSOD (%)	1.65 ± 0.14	1.49 ± 0.13	1.30 ± 0.11	1.00 ± 0.18 *^,#^
MPO (%)	5.15 ± 0.81	2.63 ± 0.24 *	1.79 ± 0.33 *^,#^	1.00 ± 0.10 *^,#^
XOX (%)	3.13 ± 0.21	1.70 ± 0.09 *	1.34 ± 0.09 *	1.00 ± 0.07 *^,#^
Oxidative damage markers				
MDA (µM)	0.11 ± 0.008	0.068 ± 0.002 *	0.058 ± 0.002 *	0.047 ± 0.002 *^,#^
Carbonyl index (%)	175 ± 25	137 ± 22	116 ± 18 *	100 ± 29 *

The effects of HBOT sessions were evaluated by a one-way ANOVA. *p* < 0.05; * indicates significant differences with respect to session 1; ^#^ indicates significant differences with respect to session 5; ^&^ indicates significant differences with respect to session 20. Abbreviations: CAT, catalase; EcSOD, extracellular superoxide dismutase; MPO, myeloperoxidase; XOX, xanthine oxidase; MDA, malondialdehyde.

## Data Availability

Researchers wishing to access the data used in this study can make a request to the corresponding author: antoni.sureda@uib.es.

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
