# Peer review of "Hyperbaric Oxygen Therapy Reduces Oxidative Stress and Inflammation, and Increases Growth Factors Favouring the Healing Process of Diabetic Wounds"

_ijms, 2023, doi:10.3390/ijms24087040_

Round 1
Reviewer 1 Report
Capó et al. explores the hematological, biochemical and inflammatory/healing marker changes that occur in patients undergoing hyperbaric oxygen therapy (HBOT) to treat chronic diabetic wounds. This study is of interest to healthcare professionals and scientists who wish to explore the changes to oxidative, inflammatory and healing factors in the blood of patients undergoing HBOT treatment in order to understand its implications and effectiveness over the course of therapy.
Major comment:
1. In the figures and text, it might be better to call the “1 month after” timepoint of observation as simply “Day 28” to keep the units consistent with the earlier timepoints that are reported in days.
2. Given that the study mainly has described the hematological, biochemical and inflammatory/healing markers in blood from the 18 HBOT patients, please add information on the therapeutic outcome if there it can be documented (i.e. how many of the 18 patients responded with better clinical outcomes?). This information may be added at the beginning paragraph of the results section.
3. Please include information of the catalog number of the antibodies used for Western blotting, for reproducibility. The western blot images may also be added to the manuscript as a supplementary file.
Minor comment:
1. In the abstract, the word “growth factors” have been miswritten as “grow” factors in line 22, 32 and 37. Please correct them to instead read “growth” factors.
2. In line 37, please change “release grow factors” to instead read “release of growth factors”.
3. Lines 58 and 64, please change diabetic “food” to diabetic “foot”.
4. Typological error in line 104 (“abd”) to be fixed.
5. In figure 1, TNF-alpha plot – please fix typological error “1 moth after” to “1 month after” (or “Day 28”).
6. In figures 1, 2 and 3, the caption sentence “& indicates significant differences respect to session 20.” may be removed since this symbol is not used in the figures.
Author Response
Reviewer 1
Capó et al. explores the hematological, biochemical and inflammatory/healing marker changes that occur in patients undergoing hyperbaric oxygen therapy (HBOT) to treat chronic diabetic wounds. This study is of interest to healthcare professionals and scientists who wish to explore the changes to oxidative, inflammatory and healing factors in the blood of patients undergoing HBOT treatment in order to understand its implications and effectiveness over the course of therapy.
Major comment:
- In the figures and text, it might be better to call the “1 month after” timepoint of observation as simply “Day 28” to keep the units consistent with the earlier timepoints that are reported in days.
In the figures and text all “1 month after” have been changed to “Day 28” or “28 days after” in the revised version.
- Given that the study mainly has described the hematological, biochemical and inflammatory/healing markers in blood from the 18 HBOT patients, please add information on the therapeutic outcome if there it can be documented (i.e. how many of the 18 patients responded with better clinical outcomes?). This information may be added at the beginning paragraph of the results section.
In accordance with the reviewer's comment, the required information has been incorporated at the beginning of the results section. The new information is as follows: "At the beginning of the study, a total of 21 patients were included, however 2 of them voluntarily decided to leave the study before its completion and 1 of them did not respond to therapy and total wound healing was not achieved after treatment with HBOT and, therefore, it was not included in the final analysis of the data”.
- Please include information of the catalog number of the antibodies used for Western blotting, for reproducibility. The western blot images may also be added to the manuscript as a supplementary file.
The catalogue number of the antibodies used have been added in the revised version of the manuscript. Also, western blot images have been added as a supplementary file in the revised version.
Minor comment:
- In the abstract, the word “growth factors” have been miswritten as “grow” factors in line 22, 32 and 37. Please correct them to instead read “growth” factors.
The term “grow” has been changed to “growth” in the revised version.
- In line 37, please change “release grow factors” to instead read “release of growth factors”.
The following “release grow factors” has been changed to “release of growth factors” in the revised version.
- Lines 58 and 64, please change diabetic “food” to diabetic “foot”.
The word “food” has been changed to “foot” in the revised version.
- Typological error in line 104 (“abd”) to be fixed.
“abd” has been changed to “and” in the revised version.
- In figure 1, TNF-alpha plot – please fix typological error “1 moth after” to “1 month after” (or “Day 28”).
In figure 1, “1 moth after” has been changed to “Day 28” in the revised version.
- In figures 1, 2 and 3, the caption sentence “& indicates significant differences respect to session 20.” may be removed since this symbol is not used in the figures.
The symbol “&” has been delated from figures 1, 2 and 3 in the revised version of the manuscript.

Reviewer 2 Report
In the manuscript entitled "Hyperbaric oxygen therapy reduces oxidative stress and inflammation and increases growth factors favouring the healing process of diabetic wounds" the authors investigated the mechanisms involved in the improvement of the diabetic foot in patients undergoing HBOT using biomarkers of oxidative stress, inflammation and 100 growth factors. The manuscript is relatively well presented, although I suggest the authors to address the following issues in the interests of clarity.
1. It is better if the authors illustrate the specific objectives of this work by adding an informative figure/scheme in the manuscript.
2. It would have been better if the authors highlight the signaling cascades and therapeutic targets in HBOT. It would have given a better impression to the readers.
3. The abbreviations and typos in the manuscript need double-checking.
4. In the manuscript, the following references may be considered:
DOI: 10.3390/biom11081210
DOI: 10.1016/j.pdpdt.2021.102697
5. It is better if the authors include more specific points of findings at the end of the manuscript as a conclusion section to reflect the overall understanding of the work.
Author Response
Reviewer 2
In the manuscript entitled "Hyperbaric oxygen therapy reduces oxidative stress and inflammation and increases growth factors favouring the healing process of diabetic wounds" the authors investigated the mechanisms involved in the improvement of the diabetic foot in patients undergoing HBOT using biomarkers of oxidative stress, inflammation and 100 growth factors. The manuscript is relatively well presented, although I suggest the authors to address the following issues in the interests of clarity.
- It is better if the authors illustrate the specific objectives of this work by adding an informative figure/scheme in the manuscript.
We have added a new figure to illustrate the specific objectives and procedure in the revised version.
- It would have been better if the authors highlight the signaling cascades and therapeutic targets in HBOT. It would have given a better impression to the readers.
In accordance with the reviewer's comment, information on possible signalling pathways involved in HBOT-induced wound healing has been added to the discussion.
To support the added information, a series of references have been also incorporated:
Johnston, B.R.; Ha, A.Y.; Brea, B.; Liu, P.Y. The Mechanism of Hyperbaric Oxygen Therapy in the Treatment of Chronic Wounds and Diabetic Foot Ulcers. Rhode Island medical journal 2016, 99(2), 26–29.
Verma R, Chopra A, Giardina C, Sabbisetti V, Smyth JA, Hightower LE, Perdrizet GA. Hyperbaric oxygen therapy (HBOT) suppresses biomarkers of cell stress and kidney injury in diabetic mice. Cell Stress Chaperones 2015 May;20(3):495-505.
Szade, A.; Grochot-Przeczek, A.; Florczyk, U.; Jozkowicz, A.; Dulak, J. Cellular and molecular mechanisms of inflammation-induced angiogenesis. IUBMB Life 2015, 67, 145–159.
Niu, C.-C.; Lin, S.-S.; Yuan, L.-J.; Lu, M.-L.; Ueng, S.W.N.; Yang, C.-Y.; Tsai, T.-T.; Lai, P.-L. Upregulation of miR-107 expression following hyperbaric oxygen treatment suppresses HMGB1/RAGE signaling in degenerated human nucleus pulposus cells. Arthritis Res Ther 2019, 21, 1–14.
Fallah, A.; Sadeghinia, A.; Kahroba, H.; Samadi, A.; Heidari, H.R.; Bradaran, B.; Zeinali, S.; Molavi, O. Therapeutic targeting of angiogenesis molecular pathways in angiogenesis-dependent diseases. Biomed Pharmacother.2019, 110, 775–785.
Sunkari, V.G., Lind, F., Botusan, I.R., Kashif, A., Liu, Z. J., Ylä‐Herttuala, S., Brismar, K., Velazquez, O., Catrina, S.B. (2015). Hyperbaric oxygen therapy activates hypoxia‐inducible factor 1 (HIF‐1), which contributes to improved wound healing in diabetic mice. Wound Repair Regen 23(1), 98-103.
- Zhang, Q.; Gould, L.J. Hyperbaric oxygen reduces matrix metalloproteinases in ischemic wounds through a redox-dependent mechanism. J Invest Dermatol 2014, 134, 237–246.
- The abbreviations and typos in the manuscript need double-checking.
The abbreviations and typos have been checked in the revised version.
- In the manuscript, the following references may be considered:
DOI: 10.3390/biom11081210
DOI: 10.1016/j.pdpdt.2021.102697
According to the reviewer comment the following references were considered and included to the text:
De Wolde, S. D., Hulskes, R. H., Weenink, R. P., Hollmann, M. W., & Van Hulst, R. A. (2021). The Effects of Hyperbaric Oxygenation on Oxidative Stress, Inflammation and Angiogenesis. Biomolecules, 11(8), 1210. DOI: 10.3390/biom11081210.
Kadkhoda, J., Tarighatnia, A., Barar, J., Aghanejad, A., & Davaran, S. (2022). Recent advances and trends in nanoparticles based photothermal and photodynamic therapy. Photodiagnosis and photodynamic therapy, 37, 102697. DOI: 10.1016/j.pdpdt.2021.102697.
- It is better if the authors include more specific points of findings at the end of the manuscript as a conclusion section to reflect the overall understanding of the work.
Additional information has been added to the conclusion section to improve the understanding of the presented work.
Round 2
Reviewer 2 Report
The authors addressed all my concerns.